# Effects of Extrusion and Irradiation on the Mechanical Properties of a Water–Collagen Solution

**DOI:** 10.3390/polym14030578

**Published:** 2022-01-31

**Authors:** Hynek Chlup, Jan Skočilas, Jaromír Štancl, Milan Houška, Rudolf Žitný

**Affiliations:** 1Department of Mechanics, Biomechanics and Mechatronics, Faculty of Mechanical Engineering, Czech Technical University in Prague, Technická 4, 160 00 Prague, Czech Republic; hynek.chlup@fs.cvut.cz; 2Department of Process Engineering, Faculty of Mechanical Engineering, Czech Technical University in Prague, Technická 4, 160 00 Prague, Czech Republic; jaromir.stancl@fs.cvut.cz (J.Š.); rudolf.zitny@fs.cvut.cz (R.Ž.); 3Food Research Institute Prague, Radiová 1285/7, 102 00 Prague, Czech Republic; milan.houska@vupp.cz

**Keywords:** collagen, anisotropy, electron irradiation, extrusion, tensile test

## Abstract

This article describes 1D extension tests on bovine collagen samples (8% collagen in water). At such a high collagen concentration, the mechanical properties of semi-solid samples can be approximated by hyperelastic models (two-parametric HGO and Misof models were used), or simply by Hooke’s law and the modulus of elasticity E. The experiments confirm a significant increase in the E-modulus of the samples irradiated with high-energy electrons. The modulus E ~ 9 kPa of non-irradiated samples increases monotonically up to E ~ 250 kPa for samples absorbing an e-beam dose of ~3300 Gy. This amplification is attributed to the formation of cross-links by irradiation. However, E-modulus can be increased not only by irradiation but also by exposure to a high strain rate. For example, soft isotropic collagen extruded through a 200 mm long capillary increases the modulus of elasticity from 9 kPa to 30 kPa, and the increase is almost isotropic. This stiffening occurs when the corrugated collagen fibers are straightened and are aligned in the flow direction. It seems that the permanent structural changes caused by extrusion mitigate the effects of the ex post applied irradiation. Irradiation of extruded samples by 3300 Gy increases the modulus of E-elasticity only three times (from 30 kPa to approximately 90 kPa). Extruded and ex post irradiated samples show slight anisotropy (the stiffness in the longitudinal direction is on an average greater than the transverse stiffness).

## 1. Introduction

Collagen-based materials are used, for example, in manufacturing vascular grafts in biomedicine [1], and in making sausage casings in the food industry [2]. However, there is little information regarding the mechanical properties of these collagenous systems. Skocilas et al. [3] used an extrusion rheometer on a fluid-like water suspension of native collagen and described the “shear” behavior at high shear rates, using a power law model. Semisolid highly elastic collagenous systems (for example tendons) are usually tested using extension machines, which provide information about the elastic modulus, stress relaxation, and the ultimate tensile strength [4]. During or in combination with uniaxial loading tests, structural information regarding the arrangement of collagen fibers can be determined using atomic force microscopy, electron microscopy, orcein staining, or X-ray scattering. This information enables entropic structural models describing the “toe” and “heel” regions of the stress–strain relationship to be identified [5,6].

The mechanical, thermodynamic, and biochemical properties of collagen can be modified through the addition of suitable additives (e.g., glutaraldehyde or polyvinyl alcohol) or through irradiation, which affects the crosslinking between long collagen macromolecules; however, in some cases the result is fragmentation of the irradiated dried collagen macromolecules. Electron irradiation [7], UV irradiation [8] and gamma irradiation [9] have all been used on various collagen materials; these tests were usually performed on soft tissues, e.g., tendons [6], or pericardium [4]. Only a small number of experiments have focused on the effects of radiation on the extracellular matrix or on water-collagen mixtures [10,11]. Riedel et al. [12] found out that, for collagen matrices (rat tail collagen, 2 mg/mL) irradiated by a dose from 0–100 kGy, the crosslinking density increases with the irradiation dose.

Water-soluble polymers can be converted into hydrogels with the use of irradiation [12]. The properties of the irradiated product depend on the concentration of the polymer and the radiation dose. In general, the crosslink density increases with the polymer concentration and the radiation dose [13]. Usually, high power of the dose is used for crosslinking enhancement from 10 kGy up to 0.5 MGy. Manaila et al. [14] applied high power of the dose of electron beam irradiation of 150, 300 and 450 kGy to investigate the mechanical properties and the degradation process of a composite of natural rubber and plasticized starch. They found that the tensile strength and also the cross-link density increase slightly at 150 kGy and then decrease with the increase in the irradiation dose.

Jamal et al. [15] investigated the effect of irradiation treatment on the mechanical properties of samples made from high-density polyethylene (HDPE, 70 vol%), ethylene propylene diene monomer rubber (EPDM 30 vol%) and nanoparticles of organophilic montmorillonite clay (OMMT, 2–8 vol%). An irradiation dose of 50, 100, 150 and 200 kGy was applied. The mechanical properties (the tensile strength) of the mixture increase with an irradiation dose of up to 100 kGy, then decrease with the irradiation dose. The best improvement of the mechanical properties was achieved with clay (4 vol%) and with a 100 kGy irradiation dose. The same results were obtained by the authors of [16] for EPDM polymer using gamma irradiation treatment (up to 1.4 MGy), and by [17] using gamma irradiation treatment (up to 250 kGy).

A similar phenomenon was observed after irradiation of chitosan powder for membrane production, see [18]. The molecular weight of a chitosan solution and its mechanical properties (Young’s modulus, tensile strength and elongation at break) decreased with the increase in the irradiation dose (12.5, 100 and 250 kGy). This behavior can be attributed to a reduction in molecular weight and degradation of the molecular chain. 

The authors of [19] used e-beam treatment of sulfonated polyether ketone (CSPEEK) membranes and investigated the mechanical properties of the membranes. They used doses in the range from 25 to 200 kGy, and they found that both the stress at break and the strain at break increased gradually with the increase in the irradiation dose, but no significant changes were observed from the yield points. The results showed that the CSPEEK membranes became harder and the degree of crosslinking increased with the increasing irradiation dose.

Similar trends [20] of dependence of mechanical properties on irradiation dose were observed for chitosan blends with polyvinyl alcohol PVA within the range of 0–75 kGy by [21,22].

Similar results were obtained for other polymer materials by [23] for silica-filled silicone elastomer (up to 200 kGy gamma irradiation treatment), by [24] for fluorocarbon elastomer (up to 200 kGy), and by [25] for styrene/butadiene-based linear triblock copolymer (up to 200 kGy).

Cui and Zhang [26] also used electron beam irradiation to modify the mechanical and thermal properties of ternary polyamide copolymer (TPA) crosslinked by various agents. They also found the maximum tensile strength of samples irradiated by doses between 100–250 kGy with respect to the crosslinking agent that was used. The range of the irradiation dose was from 50–300 kGy. The trend of the dependency of the mechanical properties on the irradiation dose is the same. There is an optimum dose corresponding to the maximum tensile strength for each polymer and for each mixture of polymer with additives. Above this optimum dose, the mechanical properties of the treated sample decrease. In order to improve or optimize the mechanical properties of a polymer product by irradiation treatment, the maximum dose has to be found, e.g., for a further engineering application [27,28].

The aim of this paper is to describe the changes in the mechanical properties of a bovine collagen solution after electron irradiation at various energy levels, which is an interesting and available method for changing the mechanical properties of biopolymer products. An even more interesting conclusion concerns the significant difference in mechanical properties in the highly concentrated isotropic collagen mass after passing through the narrow slit of the extruder may be due to structural changes in the material. 

## 2. Materials and Methods

Extension tests were performed on an aqueous solution of natural bovine collagen (type I) with a collagen content of 8.0% by weight. Bovine collagen matter is made in tanneries from cattle hides according to established techniques [29]. The hypodermis is separated from the epidermis, the fat and the hair by soaking, liming, neutralization and bleaching, and the final material is washed, cut, milled and mixed with water to the required water/collagen content. No additives were used for the material investigated here. Further information on the composition and other properties of non-irradiated collagen material has been published previously [30]. Namely, the existence of very long collagen fibers (24% of the longest fraction 780 kDa) was identified with the use of size exclusion chromatography and UV detection, see [30]. Viscoelastic behavior of the material was detected by a cone and plate oscillating rheometer, modules of *G*′ 15–18 kPa and *G*″ of 18–21 kPa were measured, see [30]. Measurements of the thermal properties of the collagen matter by DSC showed thermal capacity values in the range 3700–4400 J/(kg·K) and the transition temperature into gelatin in the range between 33–43 °C, see [31]. The parameters of the power-law model of the shear flow properties were identified as N = 0.28 and K = 700 Pa·s^n^ for the investigated collagen matter, see [3].

Isotropic samples of collagen material were irradiated at doses of 858 Gy and 3380 Gy using a Microtrone MT 25 cyclic electron accelerator; see [32]. In our experiments, an approximately constant intensity of electron irradiation was used. The distance of the irradiated samples from the e-beam emitter (500 W) was constant, and only the irradiation time changed. The cake sample was irradiated in a plastic bag, was then cut into strips (typical size 40 × 4 × 10 mm), and was clamped in the jaws of a Zwick Roell biaxial extension tester, see Figure 1. The sampling frequency of the force and displacement sensing was 20 Hz. The exact geometry and the different thicknesses of each sample were measured (optically), and the most uniform cross-section was selected for targeting the Zwick Roell optical extensometer and for evaluating the Cauchy stress. The dimensions of the isotropic samples are presented in Table 1.

The same raw collagen mass (isotropic collagen from the same batch) was used to analyze planar samples formed by extrusion of isotropic collagen through a narrow slit (cross-section 2 × 20 mm, capillary length 200 mm) at a very high strain rate of 2000 s^−1^, see [31]. The dissipated mechanical energy in the material pushed through the capillary was 2.07 ± 0.31 kJ/kg for the geometry, the process conditions (wall shear stress 11.3 kPa) and the rheology parameters (flow index 0.25 and coefficient of consistency 1700 Pa·s^n^) that were used, see [31]. The extruded strips were slightly anisotropic (when clamping the strips, it was necessary to distinguish between the longitudinal orientation and the transverse orientation of the filaments), and the orientation according to the flow direction had to be respected, see Figure 2. Otherwise, the experimental procedure was the same as in the isotropic case. The dimensions of the extruded samples are presented in Table 2.

Simple tension tests were performed. The jaw displacement was controlled during the experiment. The samples were loaded by deformation. The jaw velocity was 0.1 mm/s. The average value of the strain rate for all samples was 0.005 1/s. The force was measured using HBM U9B force sensors with a range of 50 N, accuracy class 0.5 (tensile/compressive force transducer, nominal forces 50 N, accuracy class 0.5). The recorded data were recalculated to the relationship between the deformation *ε* and the true Cauchy stress *σ*, using Matlab. Matlab functions were used for identifying the hyperelastic models and for calculating the associated statistics. The measured data are characterized by the time-independent stress stiffening and can be described using almost any hyperelastic model, e.g., three-parameter HGO [33].

For uniaxial loading and for deformation *ε* less than 0.3, the HGO model can be simplified to a two-parameter model
(1)$$\sigma =E\varepsilon +18k_{1}\varepsilon^{3}.$$

The same data can also be approximated using a two-parameter structural model [6]
(2)$$\sigma =\frac{E\varepsilon }{1-\frac{\varepsilon }{\varepsilon_{l}}}$$
where *ε_l_* is the limiting deformation. For a very small deformation *ε* (or for *k*_1_ = 0 or for *ε_l_* → ∞) both models reduce to Hooke’s model with Young’s modulus *E*.

## 3. Results and Discussion

A total of 31 samples were processed with three different doses of irradiation during our study of isotropic randomly-oriented collagen. The stress–strain relationship depends slightly on the elongation rate. However, there were no discernible trends, and therefore the viscous effects were neglected during the analysis. Measured data for three levels of irradiation (10 non-irradiated samples, 7 samples exposed to 858 Gy, and 14 samples exposed to 3380 Gy) and the corresponding predictions of the hyperelastic models are shown in Figure 3 separately for each dose. For a clear comparison, the predictions of the models are shown together in one graph, see Figure 4. 

The non-irradiated samples had the worst repeatability. For the non-irradiated samples, it was difficult to prepare a good-quality sample and the measurements were also difficult to make. The non-irradiated samples were sticky and were very pliable with minimal flexural stiffness. They were difficult to handle. To provide a clear comparison, the true (Cauchy) stress was 10 kPa for the non-irradiated samples, 110 kPa for the samples exposed to 858 Gy, and more than 270 kPa for the samples irradiated at the maximum dose of 3380 Gy, at the same 0.3 deformation. The 10 non-irradiated samples were very soft, with very high limiting deformation *ε_l_* > 1. The identified parameters of the simplified HGO model, Equation (1), are presented in Table 3, while the parameters of the Misof model, Equation (2), are presented in Table 4. Note that elastic moduli of the order of 10 kPa are quite small for non-irradiated samples, and the maximum of 170 kPa is exhibited by samples irradiated by 3380 Gy. At first glance, it seems that the HGO model produced a better approximation of the averaged data, when the standard deviations between the measured and predicted stresses are taken into consideration.

The second group of experiments investigated the effects of flow and anisotropy. Samples in the form of extruded bands were analyzed separately in the longitudinal direction (20 samples) and in the transversal direction (20 samples) at four different levels of dosing: 0, 250 Gy, 850 Gy and 3300 Gy. For each radiation dose in each direction, 5 valid measurements were evaluated. The identified model parameters are presented in Table 5 —HGO longitudinal, in Table 6—transversal, in Table 7—Misof longitudinal, and in Table 8—transversal. The measured data for longitudinal deformation are shown in Figure 5, and the transversal deformation is shown in Figure 6. For a clear comparison, the predictions of the models are shown together in a single graph, see Figure 7a for the longitudinal direction and Figure 7b for the transversal direction. For the transversal samples, the data are thinner because the samples were shorter, but the loading rate was the same as for the longitudinal samples.

Table 5, Table 6, Table 7 and Table 8 and associated Figure 7 demonstrate a significant change of the mechanical properties of collagen after soft isotropic collagen matter has passed through a narrow gap. Extruding fresh collagen through a 200 mm long capillary increases the modulus of elasticity *E* approximately threefold (from 9 kPa to 30 kPa), and the increase is almost isotropic. The increase in stiffness due to extrusion can be explained by stretching of the wavy collagen fibers flowing through a narrow slit at a very high rate of deformation. Even greater stiffening can be achieved by e-beam irradiation of extruded samples (one order of magnitude can be accomplished by a radiation dose of the order of kGy). However, the supposedly irreversible structural changes caused by extrusion mitigate the effect of ex post applied irradiation (compare the increase of *E* at the same dose of 3300 Gy for an isotropic sample and for an extruded sample). The increases in stiffness when e-bean irradiation is applied in the longitudinal direction and also in the transversal direction can be explained by the assumption of the possible formation of crosslinks in the native collagen structure and also in the irreversibly modified collagen structure. The observed slight decrease in longitudinal stiffness for a very high dose (3300 Gy) can be explained by breaking of the collagen fibers, manifested by a decrease in stiffness and in molecular mass at an extremely high intensity of radiation. The same phenomenon of stiffness decreasing with an increasing irradiation dose was observed at 10 kGy [14] and at 12.5 kGy [18], respectively. After reaching the turning point of the irradiation dose, the mechanical properties of the sample decrease due to the reduction in molecular weight and degradation of its molecular chain [18]. For a bovine collagen solution, this critical irradiation dose together with pretreatment by extrusion is approximately 3 kGy, while for untreated collagen solution the critical dose is much more than 3 kGy.

## 4. Conclusions

The stiffening of a highly concentrated collagen solution can be explained as a manifestation of significant structural changes (not measured), probably the formation of the cross-links initiated by e-beam irradiation and irreversible straightening of the collagen fibers in the flow of isotropic collagen through a narrow gap. The extruded material with fibers oriented in the direction of flow is non-isotropic, and is stiffer than untreated isotropic collagen. The results of our study have confirmed that irradiating a water-collagen suspension with an electron beam significantly increases the final stiffness. The experimentally determined deformation-stress relationship has been approximated using two parametric hyperelastic models (HGO and Misof). A significantly increased modulus of elasticity (*E, k*_1_) was found after electron irradiation (doses of 0, 250, 850 and 3300 Gy were tested). Further experiments should be carried out (on the role of concentration, critical loading and the duration of structural changes). Our results show that modification of the mechanical properties of collagen by irradiation has potential, for example, in the field of tissue engineering in addition to the food and pharmaceutical industries.

## Figures and Tables

**Figure 1 polymers-14-00578-f001:**
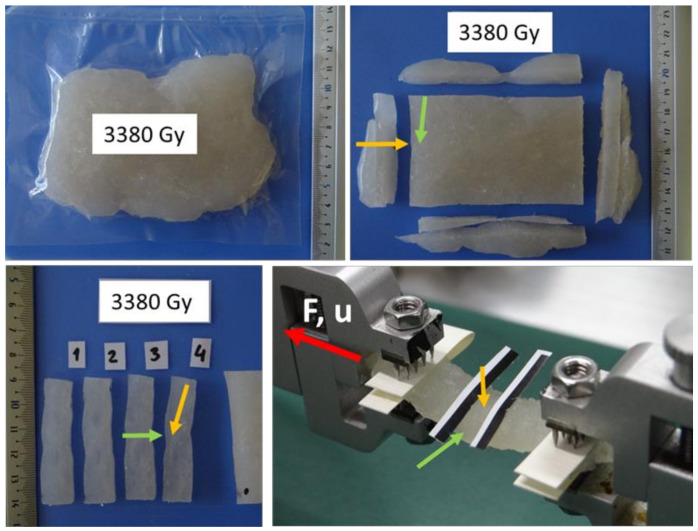
Sample storage, preparation, and attachment to a Zwick Roell biaxial extension tester. Non-extruded collagen irradiated by dose 3380 Gy. The arrows indicate the direction of sample cutting and attachment for isotropy analysis. F—force, u—displacement.

**Figure 2 polymers-14-00578-f002:**
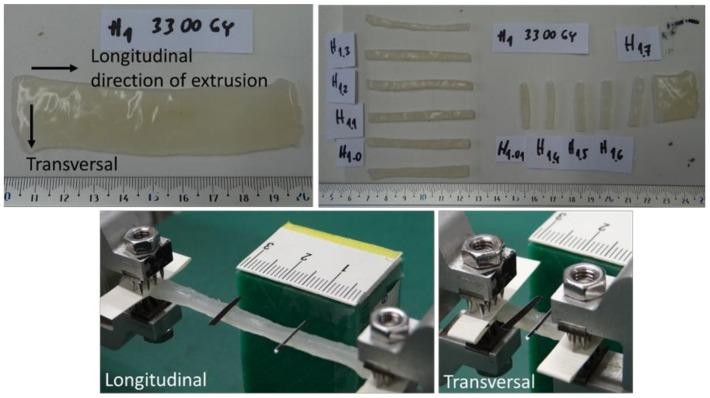
Sample preparation and attachment to a Zwick Roell biaxial extension tester. Extruded collagen irradiated by dose 3300 Gy. The arrows indicate the direction of sample orientation during extrusion.

**Figure 3 polymers-14-00578-f003:**
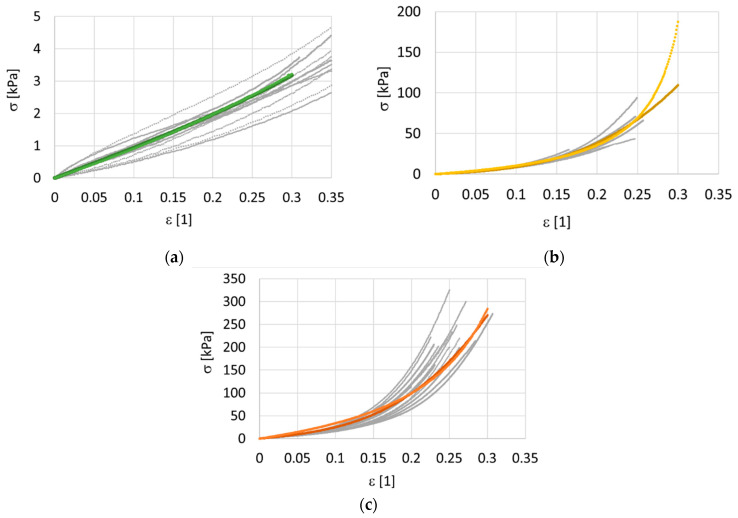
Cauchy stress—deformation data for isotropic collagen mass. Modified Holzapfel–Gasser–Ogden (HGO) model and Misof model. The grey points represent data, the colored lines represent models. Two doses of irradiation were used, 858 Gy and 3380 Gy. (**a**) non-irradiated 0 Gy, models are overlapping, (**b**) dose 858 Gy, the light line represents the HGO model, and the dark line represents the Misof model, (**c**) dose 3380 Gy, models are overlapping.

**Figure 4 polymers-14-00578-f004:**
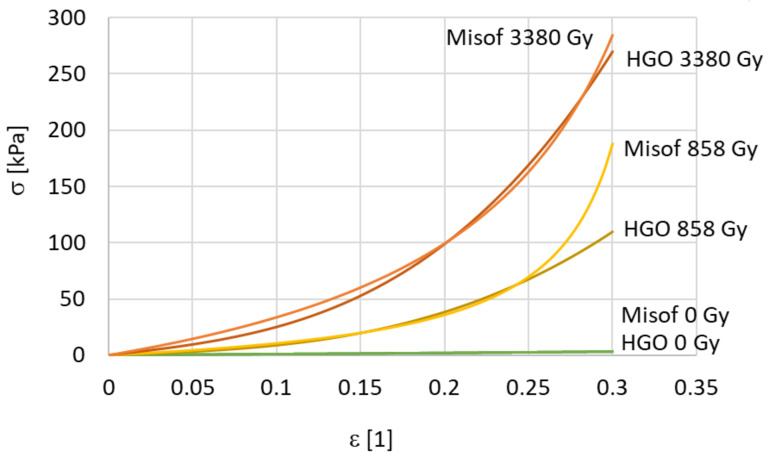
A comparison of the models for different irradiation doses, isotropic collagen mass.

**Figure 5 polymers-14-00578-f005:**
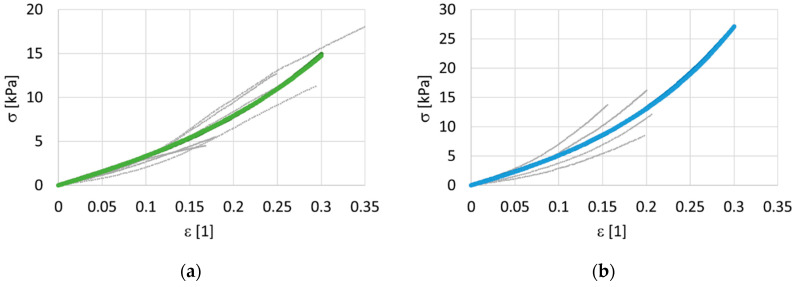
Cauchy stress—deformation for extruded collagen mass. The modified Holzapfel–Gasser–Ogden (HGO) model and the Misof model. Longitudinal direction. Three doses of irradiation were used: 250 Gy, 850 Gy and 3300 Gy. The grey points represent data, and the colored lines represent models. (**a**) non-irradiated 0 Gy, models are overlapping, (**b**) dose 250 Gy, models are overlapping, (**c**) dose 850 Gy, models are overlapping, (**d**) dose 3380 Gy, the light line represents the HGO model, and the dark line represents the Misof model.

**Figure 6 polymers-14-00578-f006:**
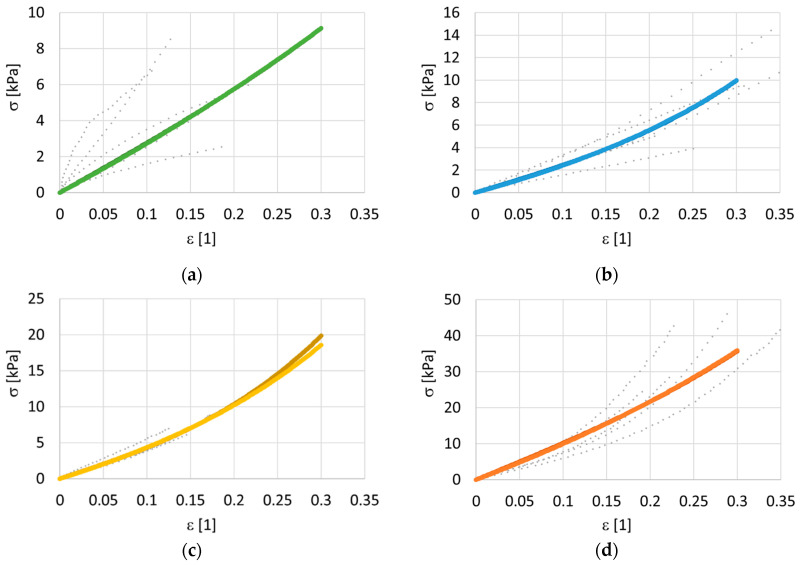
Cauchy stress—deformation for extruded collagen mass. The modified Holzapfel–Gasser–Ogden (HGO) model and the Misof model. Transversal direction. Three doses of irradiation were used: 250 Gy, 850 Gy and 3300 Gy. The grey points represent data, and the colored lines represent models. (**a**) non-irradiated 0 Gy, models are overlapping, (**b**) dose 250 Gy, models are overlapping, (**c**) dose 850 Gy, the light line represents the HGO model, and the dark line represents the Misof model, (**d**) dose 3380 Gy, models are overlapping.

**Figure 7 polymers-14-00578-f007:**
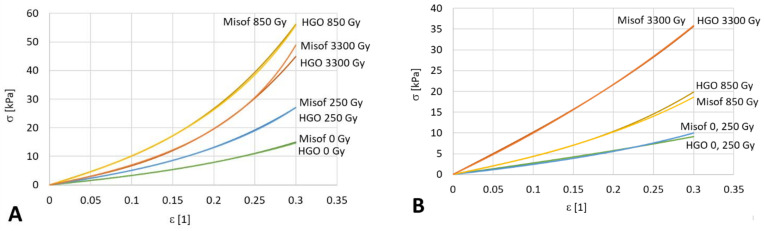
A comparison of the HGO and Misof model trends for extruded collagen mass. (**A**)—Longitudinal direction, (**B**)—Transversal direction.

**Table 1 polymers-14-00578-t001:** Dimensions of non-extruded isotropic samples. *H*—sample thickness, *B*—sample width, *A*—sample cross-section. Values with 95% confidence bounds.

Dose (Gy)	*H* (mm)	*B* (mm)	*A* (mm^2^)
0	8.27 ± 1.40	10.70 ± 1.46	89.10 ± 21.58
858	6.12 ± 0.62	11.23 ± 0.65	68.79 ± 8.33
3380	3.47 ± 0.40	10.44 ± 0.44	35.34 ± 4.63

**Table 2 polymers-14-00578-t002:** Dimensions of the extruded samples. *H*—sample thickness, *B*—sample width, *A*—sample cross section. Values with 95% confidence bounds.

Dose (Gy)	*H* (mm)	*B* (mm)	*A* (mm^2^)
0	2.18 ± 0.12	6.25 ± 0.37	13.61 ± 0.97
250	2.10 ± 0.06	4.60 ± 0.28	9.66 ± 0.59
850	2.17 ± 0.11	3.98 ± 0.19	8.63 ± 0.42
3300	2.20 ± 0.12	4.08 ± 0.24	8.98 ± 0.54

**Table 3 polymers-14-00578-t003:** Isotropic samples. Parameters of the simplified HGO model. Values with 95% confidence bounds.

Dose (Gy)	*E* (kPa)	*k*_1_ (kPa)	R-Square (-)
0	9.24 ± 0.10	0.84 ± 0.02	0.94
858	52.55 ± 1.96	193.20 ± 2.70	0.96
3300	169.9 ± 6.90	450.9 ± 7.70	0.89

**Table 4 polymers-14-00578-t004:** Isotropic samples. Parameters of the Misof model. Values with 95% confidence bounds.

Dose (Gy)	*E* (kPa)	*ε_l_* (-)	R-Square (-)
0	8.49 ± 0.08	1.439 ± 0.022	0.94
858	73.55 ± 0.95	0.337 ± 0.002	0.95
3380	254.4 ± 3.60	0.414 ± 0.003	0.87

**Table 5 polymers-14-00578-t005:** Extruded samples. Parameters of the simplified HGO model. Longitudinal direction. Values with 95% confidence bounds.

Dose (Gy)	*E* (kPa)	*k*_1_ (kPa)	R-Square (-)
0	30.96 ± 0.83	11.58 ± 0.74	0.95
250	46.00 ± 3.65	27.31 ± 7.75	0.79
850	90.16 ± 3.50	59.85 ± 10.02	0.90
3300	56.88 ± 2.00	57.35 ± 3.51	0.96

**Table 6 polymers-14-00578-t006:** Extruded samples. Parameters of the simplified HGO model. Transversal direction. Values with 95% confidence bounds.

Dose (Gy)	*E* (kPa)	*k*_1_ (kPa)	R-Square (-)
0	27.37 ± 0.80	1.89 ± 0.93	0.90
250	23.31 ± 2.09	6.10 ± 1.48	0.89
850	40.39 ± 1.64	15.90 ± 4.46	0.97
3300	99.82 ± 5.89	11.85 ± 3.45	0.90

**Table 7 polymers-14-00578-t007:** Extruded samples. Parameters of the Misof model. Longitudinal direction. Values with 95% confidence bounds.

Dose (Gy)	*E* (kPa)	*ε_l_* (-)	R-Square (-)
0	28.64 ± 0.73	0.718 ± 0.031	0.95
250	41.95 ± 3.78	0.559 ± 0.121	0.79
850	82.78 ± 3.72	0.538 ± 0.067	0.90
3300	54.30 ± 2.03	0.452 ± 0.016	0.96

**Table 8 polymers-14-00578-t008:** Extruded samples. Parameters of the Misof model. Transversal direction. Values with 95% confidence bounds.

Dose (Gy)	*E* (kPa)	*ε_l_* (-)	R-Square (-)
0	25.96 ± 1.11	2.050 ± 0.774	0.90
250	21.09 ± 1.92	0.824 ± 0.136	0.89
850	37.80 ± 2.06	0.771 ± 0.187	0.97
3300	91.85 ± 6.56	1.286 ± 0.289	0.90

## Data Availability

Exclude this statement. All important data are reported within article.

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
