# Peer review of "Effects of Extrusion and Irradiation on the Mechanical Properties of a Water–Collagen Solution"

_polymers, 2022, doi:10.3390/polym14030578_

Round 1
Reviewer 1 Report
The paper "Effect of extrusion and irradiation on the mechanical properties of a water-collagen solution" by Hynek Chlup and co. presents the changes in mechanical properties of bovine collagen solution after electron irradiation at different energy levels.
The subject is interesting, but some of the conclusions are not supported by the presented results, as follows:
Pag 2 row 66 "An even more interesting conclusion concerns the significant structural changes in the highly concentrated isotropic collagen mass after passing through the narrow slit of the extruder."
Pag 9 row 197 : "On the other hand, the irreversible structural changes caused by the extrusion mitigate the effect of ex post applied irradiation (compare the increase of E at the same dose 3300 Gy for isotropic and for the extruded sample). Stiffness increases by e-beam irradiation in the longitudinal as well as transversal directions can be ascribed to the formation of crosslinks in the native as well as in the irreversibly modified collagen structure. Observed slight decrease of longitudinal stiffness at a very high dose (3300 Gy) can be explained by breaking collagen fibers, manifested by decrease of stiffness and molecular mass at extremely high intensity of radiation."
Pag 9 row 212 "Stiffening of a highly concentrated collagen solution is a manifestation of significant structural changes, probably formation of the cross-links initiated by e-beam irradiation and irreversible straightening of collagen fibers in the flow of isotropic collagen through a narrow gap."
Usually, for explain a structural change in collagen structure, a second characterisation method (FTIR as example) is needed, otherwise the above conclusions are only assumptions.
Reviewer 2 Report
The analysis of the literature on the topic needs to be expanded to include more literature items.
The manufacturing process for the test materials, including parameters, should be fully described
The article presents only selected mechanical properties.
The analysis should be extended to include comprehensive tests of mechanical properties.
It would also be advisable to investigate the mechanical properties by the DMTA method and the thermal properties by the differential scanning calorimetry method.
Round 2
Reviewer 1 Report
The paper was improved